# Machine Learning for Detection of Macroalgal Blooms in the Mar Menor Coastal Lagoon Using Sentinel-2

Encarni Medina-López [1], Gabriel Navarro [2], Juan Santos-Echeandía [3], Patricia Bernárdez [3] and Isabel Caballero [2],*

1    Institute for Infrastructure and Environment, School of Engineering, The University of Edinburgh, Edinburgh EH9 3FG, UK
2    Institute of Marine Sciences of Andalusia (ICMAN-CSIC), 11519 Puerto Real, Spain
3    Vigo Oceanographic Centre, Spanish Oceanographic Institute (IEO-CSIC), 36390 Vigo, Spain
*    Correspondence: isabel.caballero@icman.csic.es

**Abstract:** The Mar Menor coastal lagoon in southeastern Spain has experienced a decline in water quality due to increased nutrient input, leading to the eutrophication of the lagoon and the occurrence of microalgal and macroalgal blooms. This study analyzes the macroalgal bloom that occurred in the lagoon during the spring-summer of 2022. A set of machine learning techniques are applied to Sentinel-2 satellite imagery in order to obtain indicators of the presence of macroalgae in specific locations within the lagoon. This is supported by in situ observations of the blooming process in different areas of the Mar Menor. Our methodology successfully identifies the macroalgal bloom locations (accuracies above 98%, and Matthew's Correlation Coefficients above 78% in all cases), and provides a probabilistic approach to understand the likelihood of occurrence of this event in given pixels. The analysis also identifies the key parameters contributing to the classification of pixels as algae, which could be used to develop future algorithms for detecting macroalgal blooms. This information can be used by environmental managers to implement early warning and mitigation strategies to prevent water quality deterioration in the lagoon. The usefulness of satellite observations for ecological and crisis management at local and regional scales is also highlighted.

**Keywords:** Mar Menor coastal lagoon; Copernicus program; macroalgal bloom; K-Means; CART; machine learning

## 1. Introduction

The Mar Menor coastal lagoon is the largest hypersaline lagoon in Europe (Figure 1). This lagoon covers an area of 170 km$^2$ and a perimeter of 59.51 km. It has a mean depth of 3.6 m and a maximum depth of 6 m [1]. "La Manga", a 22 km long and 100–900 m wide sandbar, acts as a barrier between the lagoon and the Mediterranean Sea. On the one hand, this barrier is crossed by five inlets called golas that determine the water exchange with the Mediterranean Sea. The opening of these channels caused the major changes recorded for the lagoon dynamics [1]. On the other hand, in the western part of the lagoon, there are several wadis that contribute water and materials from agricultural and mining run-off and with high nutrient load, mainly the Albujón watercourse, which is the only permanent wadi in the area [1–3]. Evaporation exceeds rainfall and run-off and, therefore, the salinity normally ranges from 41 to 46. Most of the watercourses discharge in the southern half of the lagoon, depending on the sporadic and torrential rainfall regime, which occurred in the autumn of 2019 during one of the most extreme storm events in the area [4].

The Mar Menor is a place of vital natural importance where up to 10 approved environmental protection figures and other catalogs of geological and ecosystem interest come together. For example, it is a protected area under Natura 2000 "MAR MENOR", classified as a Special Conservation Zone and Special Protection Area (SPA) for Birds

(ZEPA). Within the SPA is the Protected Landscape "Open Spaces and Mar Menor Islands". In addition, the Mar Menor is a Wetland of International Importance (HII), in accordance with the Convention on Wetlands of International Importance (Ramsar Convention), and is a Specially Protected Area of Mediterranean Importance (SPAMI), in the list of the Barcelone Convention. The Mar Menor and its associated wetlands are also a Wildlife Protection Area (Law 7/1995, 21 April). Despite the natural importance of the Mar Menor lagoon and its surroundings, during recent decades, the severe degradation of the lagoon has mainly been caused by the eutrophication process due to the excess of nutrients coming from the surrounding agriculture practices [2], and references therein. This has caused a degradation of the lagoon during several periods (summer of 2015 and early 2016) [5]. This degradation has been aggravated by the high levels of nutrients and sediments in the runoff caused by the intense storm called "cold drop", which had catastrophic impacts on this area in the second week of September 2019 [4]. The event caused an increment in nitrate concentrations of about 13 mg/m$^3$, one hundred times above normal values [2]. This massive input of nutrients together with the load that was already supporting the lagoon resulted in a "green soup" and the consequent anoxia process, causing the system to collapse [2]. This event, together with the phytoplanktonic bloom that occurred during August 2021 up to a chlorophyll concentration of 20 mg/m$^3$ in some areas, caused massive mortality of several species inhabiting the lagoon, mainly fishes [6].

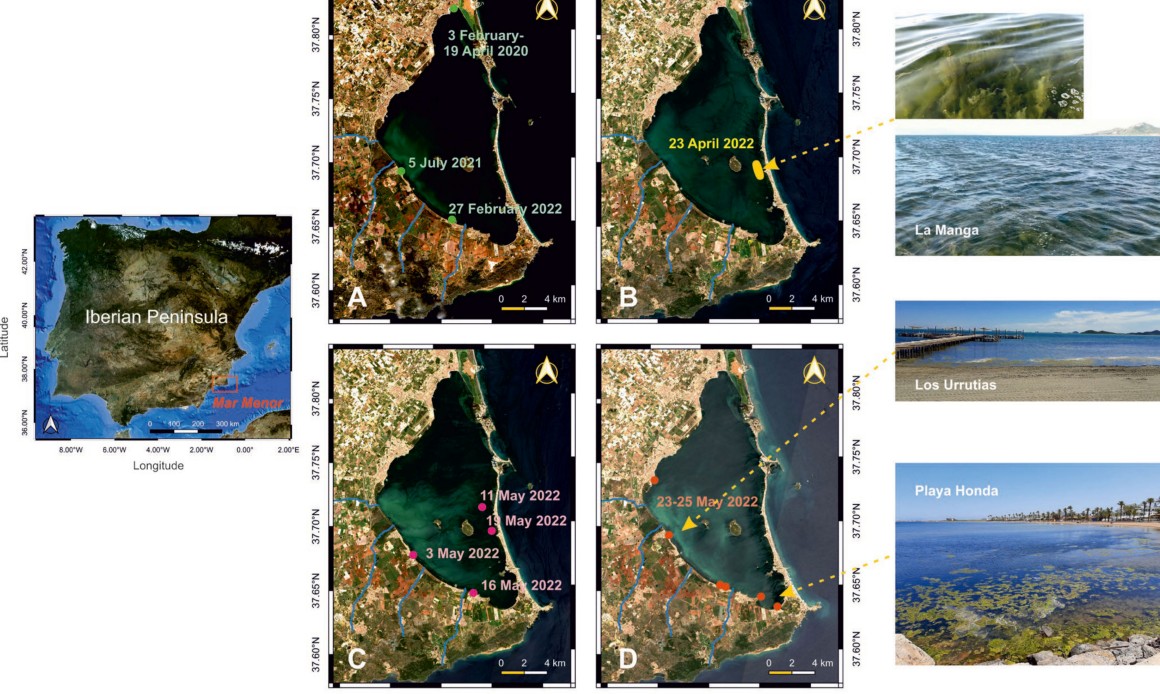

**Figure 1.** Map of the Iberian Peninsula. The red box delimits the Mar Menor coastal lagoon. In situ locations (including photographs) where macroalgae were observed by the research team on different dates from 2020 to 2022. RGB composite images (true color image Level 2A, 10 m spatial resolution) corresponding to (**A**) 24 February, (**B**) 25 April, (**C**) 10 May, and (**D**) 25 May 2022 are shown.

The eutrophication processes and the system collapse also resulted in increased media attention and the different administrations and public authorities with competencies in the management of the lagoon have been developing initiatives to try to solve the problem. The Regional Government maintains a network for monitoring variables related to water quality, such as temperature, salinity, chlorophyll, turbidity, water clarity, and dissolved oxygen. These processed data are open-access and available online (https://marmenor.upct.es/, accessed on 1 September 2022). However, these data are spatially scarce and, therefore,

there is a lack of spatial distribution. The lagoon average chlorophyll data from U.P. Cartagena in 2022 (https://marmenor.upct.es/, accessed on 1 September 2022) has been used as general guidance in this study. Remote sensing images can be a powerful complementary tool for coastal managers to be able to monitor the water quality of the lagoon in a synoptic way and quasi-real-time. In fact, there are very recent studies that used remote sensing techniques to monitor the water quality inside the lagoon [4,5,7–10]. During 2022, the Mar Menor lagoon experienced an intense proliferation of macroalgae blooms in several parts of the lagoon (these blooms are called "ovas" by the local population, Figure 1) in response to the high nutrient input. The predominant species, Chaetomorpha linum (O.F. Müller) Kützing 1845, has been part of the ecosystem since several years ago, but its extension was limited to some shoreline areas of the Mar Menor. However, in 2022, more than 17,000 Tons of this macroalgae were removed from the lagoon in 8 months (Murcia Regional Government and Mar Menor Fishermen's Guild communication). This represents an amount ten times higher than what was removed in the previous four years (1600–1700 Tons per year). These macroalgae usually grow in the presence of high concentrations of nutrients, so their removal from the lagoon helps to reduce the concentration of these nutrients. They also grow in calm shallow and protected areas, with high salinity and temperature in the presence of seagrass meadows [11], especially in the northern areas of the lagoon (https://canalmarmenor.carm.es/inventario-ecologico/flora/alga-valonia-valonia-utricularis-2/, accessed on 1 September 2022). Another problem caused by the proliferation of macroalgae is that they prevent fishing as they adhere to the fishing nets, which represents a great economic loss for the fishermen.

Several remote sensing approaches were recently carried out to monitor the algal bloom distribution over coastal regions [12–16]. In addition, remote sensing approaches have been developed during recent decades to monitor the water quality in inland and coastal waters [17,18]. Many studies have used images from the Sentinel-2 satellite. This satellite is an Earth Observation mission by the European Union's Copernicus Program that systematically acquires optical imagery at high spatial resolution. The mission is a constellation of two twin satellites launched in June 2015 (Sentinel-2A) and March 2017 (Sentinel-2B). The temporal resolution of this mission is five days at the equator and it includes 13 bands in the visible, near-infrared, and shortwave infrared parts of the spectrum. The spatial resolutions of these bands are 10, 20, and 60 m. Recently, images from this satellite processed using ACOLITE atmospheric correction were used for the quantification of sediments inside the lagoon following an extreme storm in September 2019 [4]; the process of downloading and processing these images was completed in several hours per tile. Each Sentinel-2 image cropped around the lagoon is several gigabytes in size and contains more than 1.5 million pixels.

Therefore, the aim of the present study is to develop a useful tool to localize in near-real-time the areas where these macroalgae accumulate. This is important to advise local and national administrations to remove the algae, avoiding further degradation of the ecosystem and allowing society to make use of its assets, such as leisure activities or fishing. The paper contains a first Section 2, which describes the in situ information used in this study, satellite data and processing, as well as the different machine learning techniques applied. The Section 3 is split into chlorophyll analysis, clustering (using the k-means algorithm) and classification results (using Classification and Regression Trees), SHapley Additive exPlanations (SHAP) values for algorithm explainability, and probabilistic analysis as a tool for environmental management and future predictions. Finally, some conclusions are presented.

## 2. Materials and Methods

### 2.1. In Situ Information

The in situ locations of the macroalgal bloom are presented in Figure 1. In situ data were used to verify and validate the measurement of the remote sensing data collected. Macroalgal blooms were identified visually either by walking along the Mar Menor shore-

line or using a pneumatic boat to explore the inner lagoon areas during three sampling campaigns carried out in July 2021, February 2022, and May 2022 (Table 1). The areas of macroalgae occurrence were geopositioned using a handheld GPS (Garmin Oregon 450T). Moreover, members of the Mar Menor Fishermen's Guild as well as the Mar Menor authorities also provided the research team with information about the location of the macroalgae accumulation and the areas of removal on different dates (Figure 1), during face-to-face meetings carried out in 2022. Finally, more information was collected from the written press detailing the days and areas of macroalgae visualizations.

**Table 1.** Summary of the in situ data with the date and the location of the macroalgae blooms.

| Date | Latitude (N) | Longitude | Source |
|---|---|---|---|
| 3 February 2020 | 37.818518 | −0.780308 | Newspapers |
| 14 February 2020 | 37.818518 | −0.780308 | Newspapers |
| 25 May 2020 | 37.818518 | −0.780308 | Fishermen |
| 2 April 2020 | 37.818518 | −0.780308 | Newspapers |
| 19 April 2020 | 37.818518 | −0.780308 | Fishermen |
| 5 July 2021 | 37.692149 | −0.835999 | Sampling campaign |
| 27 February 2022 | 37.653302 | −0.787118 | Sampling campaign |
| 23 April 2022 | 37.695776 | −0.749627 | Fishermen |
| 23 April 2022 | 37.686675 | −0.746365 | Newspapers |
| 3 May 2022 | 37.676429 | −0.824669 | Sampling campaign |
| 11 May 2022 | 37.712459 | −0.7555 | Sampling campaign |
| 16 May 2022 | 37.645719 | −0.766202 | Sampling campaign |
| 19 May 2022 | 37.693689 | −0.746672 | Sampling campaign |
| 23 May 2022 | 37.642472 | −0.746724 | Sampling campaign |
| 25 May 2022 | 37.652301 | −0.786721 | Sampling campaign |
| 25 May 2022 | 37.650738 | −0.781142 | Sampling campaign |
| 25 May 2022 | 37.692027 | −0.835934 | Sampling campaign |
| 25 May 2022 | 37.634358 | −0.730684 | Sampling campaign |

### 2.2. Sentinel-2 Satellite Data Preprocessing

Sentinel-2 imagery covering the Mar Menor lagoon on 25 April, 10 and 25 May, and 4 June 2022 was downloaded (https://scihub.copernicus.eu/, accessed on 1 September 2022) and processed. The preprocessing of the satellite imagery followed the same procedure as used by Caballero et al. [10]. The initial imagery was Top of Atmosphere (TOA) Level 1C, radiometrically and geometrically corrected. Bottom-of-Atmosphere (BOA) images were produced using ACOLITE, a very common atmospheric correction software. Sunglint correction was then performed on the images. Different products for biogeochemical monitoring were produced from the imagery, including OC3, chl_mishra, and Normalized Difference Chlorophyll Index (NDCI). OC3 is a band ratio algorithm that allows the estimation of seawater concentrations of chlorophyll-a using three bands near the blue, green, and red [19]. NDCI allows mapping of chlorophyll-a (chl_mishra) in estuarine and coastal turbid waters [20]. It is advantageous compared to the previous indices, as it provides a normalized estimate of chlorophyll and is particularly important for detecting algal blooms where ground truth data are not available for qualitative studies. OC3, chl_mishra, and NDCI are computed for every pixel of the images and used as additional bands during the machine learning algorithm training.

### 2.3. Machine Learning Algorithms

A set of different machine learning algorithms have been applied to the satellite data, taking into account the in situ information. The ultimate goal is to automatically detect the macroalgal bloom using only satellite information, and the in situ data are used to train these algorithms. Figure 2 presents a flow diagram of the algorithm as a visual explanation of the pseudocode. As mentioned in previous sections, obtaining accurate in situ information about the location of macroalgal blooms is a difficult process, and

we only have general guidance on areas where the bloom appeared (rather than specific coordinates). However, in order to obtain an accurate algorithm to classify the pixels of a satellite image as an algal bloom, a large dataset of accurate locations is needed. In order to bridge this gap, we applied a two-step approach: first, using an unsupervised algorithm over the satellite imagery to cluster the water pixels. The clustering exercise is fed back to an expert on in situ locations of algal blooms, who identifies which one of the clusters identified by the algorithm is a good match for the algal bloom. This information is taken into account and the algorithm is re-run until a suitable cluster is found that represents the algal blooms. The clusters do not provide information on what characteristics bring them together, but they are a useful starting point for investigating commonalities in different areas of the same water body. This is a key first step before applying a classification algorithm, which then proceeds to the introduction of explainability algorithms (SHAP values, as presented in the following sections) to understand what characteristics are key for identifying algal blooms.

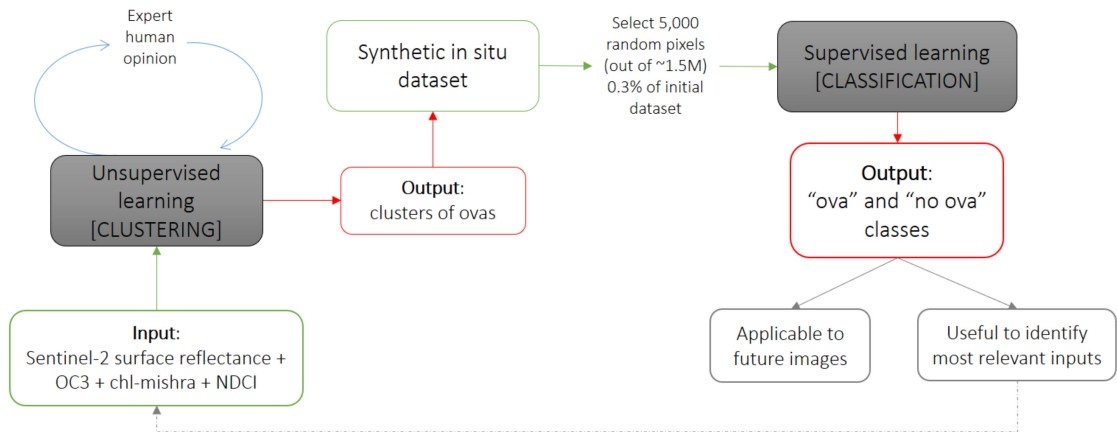

**Figure 2.** Flow diagram of the algorithm pseudocode.

Each pixel of the image used for the clustering analysis contains 16 layers of information: the 13 Sentinel-2 surface reflectance bands, and three chlorophyll and algal indices (OC3, chl_mishra, and NDCI). After the clustering analysis, we added the cluster information as an extra layer (1 = algal bloom, 0 = no algal bloom). This represents the new synthetic in situ dataset that was used in the next step. In order to avoid overfitting, and to ensure robustness, only 5000 random pixels were extracted from each image to train the classification algorithm. This represents about 0.3% of the total amount of pixels of each image. Those 5000 pixels were subsequently used to classify the image, applying a supervised algorithm. This means both input and "real" outputs must be provided to the algorithm. The algorithm then looks for relationships between inputs and outputs that are generalizable to the rest of the image (99.7% of pixels not used). In this case, the outputs are the 0–1 clustering label (with 0 being "no macroalgae", or no algal bloom present, and 1 being "macroalgae" or algal bloom present in that pixel). The result is a classification that provides both an indication of the presence of algal bloom in each pixel and its likelihood in terms of probability. This is generalizable to pixels not seen by the algorithm, and also useful for extracting information on input importance via SHapley Additive exPlanations (SHAP) values, as described in subsequent sections. The aim of developing a classification algorithm that can be explained is, on the one hand, the possibility to apply this to future images as a tool for the management of the lagoon and, on the other hand, as a tool for other semi-analytical algorithms that are not based on machine learning. Details of the algorithms tested are provided in the following sections. The code for this project was developed in Python, using the algorithms provided by the University of Waikato's WEKA project (https://www.cs.waikato.ac.nz/ml/weka/Witten_et_al_2016_appendix.pdf, accessed on 1 September 2022) via Google Earth Engine [21] locally. The WEKA project is a collection

of machine learning algorithms for data mining tasks, providing different tools for data processing, exploration, and visualization. Google Earth Engine, which includes the tools provided by WEKA, is a geospatial platform that provides a comprehensive catalog of satellite imagery and different analysis techniques using the supercomputing facilities provided by Google.

### 2.3.1. Clustering

The main algorithms that were tested in this case were K-Means and Cascade simple K-Means. K-Means is a well-known and stable clustering algorithm, first developed by Lloyd in 1982 [22]. "K" represents the number of clusters we are looking for, and it relates to the number of centroids in the data. These centroids can be physical locations or related to probabilistic measures of similarity between information in different pixels. Each point in the image is associated with a cluster, with the goal of minimizing the Euclid distance of the pixel data. The algorithm will then identify the "K" number of clusters by keeping the centroids as far apart from each other as possible. The downside of this method is that as the number of clusters is provided by the user, the process of finding the optimal number of clusters becomes iterative and may be subject to errors. The Cascade Simple K-Means algorithm was created to overcome this issue. In this case, the algorithm automatically selects the optimal number of clusters based on the Calinski–Harabasz criterion (or variance ratio criterion, VRC): well-defined clusters will present large between-cluster variance and small within-cluster variance. The optimal number of clusters corresponds to the solution with the highest Calinski–Harabasz index value [23]. This algorithm found that the optimal number of clusters in every case was always 9.

### 2.3.2. Classification

Both Support Vector Machine (SVM) and Classification and Regression Tree (CART) were tested on the data. The SVM provided consistently poorer results than the CART, and, thus, this last one was selected. A CART is a predictive model that explains how certain variables can be predicted based on other values [24]. These can be used both for regression and classification problems, although in this paper, they were used for classification. The working principle behind a CART is simple but powerful: a binary tree. Each root node in the tree represents an input variable and a split point on that variable if the variable is numeric. The leaf nodes contain the output variables, which are the target that we are aiming to achieve with our prediction. The CART will choose optimal splitting points so that a suitable tree is constructed to be able to predict outputs from inputs.

### 2.3.3. Evaluation Metrics

A set of metrics are used to evaluate the result of the classification exercise. In the equations below, TP is True Positive—actual positives that are correctly predicted; TN is True Negative—actual negatives that are correctly predicted; FP is False Positive—actual negatives that are wrongly predicted as positives; FN is False Negative—actual positives that are wrongly predicted as negatives. "Positive" in this paper represents the "macroalgae" class, while the "no macroalgae" class is "Negative".

Accuracy: represents the ratio between correctly predicted instances and all instances in the predicted dataset.

$$\text{acc} = (\text{TP} + \text{TN})/(\text{TP} + \text{TN} + \text{FP} + \text{FN})$$

The accuracy on its own is not a complete performance indicator of the classification, as it fails to provide fair estimates in class-unbalanced datasets, i.e., if one class has much more data than another class [25]. As we have much fewer data in the "macroalgae" class, we decided to include the following metrics for further checks of the goodness of fit of our algorithm.

Matthews Correlation Coefficient (MCC): This is a contingency matrix method to calculate the Pearson product-moment correlation coefficient between actual and predicted

values [26]. The MCC is only high if the predictor is able to correctly predict the majority of positive data instances and the majority of negative data instances [27].

$$MCC = (TP.TN - FP.FN)/\sqrt{((TP + FP).(TP + FN).(TN + FP).(TN + FN))}$$

Confusion matrix: The confusion matrix allows visualization of the performance of an algorithm, or the ways in which the classification model is "confused" when it makes predictions. It provides insight not only into the errors being made by the classifier but, more importantly, the types of errors that are being made. Each row of the matrix represents the instances in an actual class while each column represents the instances in a predicted class, or vice versa, distinguishing the number of True Positives, True Negatives, False Positives, and False Negatives [28].

### 3. Results and Discussion

*3.1. Chlorophyll Analysis in Areas of Interest*

Chlorophyll estimates were obtained using OC3, as described in previous sections, and used as guidance to see if the areas indicating high chlorophyll concentration matched areas where macroalgae were observed during in situ campaigns. Figure 3 presents an OC3 chlorophyll map from 10 May 2022, together with extracted transects in areas of interest. The color legend in this figure was set to a minimum of 1.7 mg/m$^3$. This was the average in situ value recorded on that day in the lagoon [29]. However, based on satellite data and the OC3 algorithm, the average value in the lagoon that day was 2.27 mg/m$^3$, and the median value was 2.52 mg/m$^3$. On the one hand, this could be due to the presence of macroalgae not being recorded by the in situ measurements of chlorophyll. On the other hand, the OC3 algorithm may be identifying the color change provided by the presence of macroalgae as chlorophyll. Most of the minimum values observed from OC3 are above the average in situ value.

The values provided by OC3 show areas with a very high concentration of chlorophyll. Three transects were measured in some of these areas: a northwest transect parallel to the coast of Los Alcázares; an east transect parallel to the coast of La Manga; a south transect parallel to the coast east of playa Honda (Figure 3). Specific chlorophyll values were plotted for these areas. In all three cases, there is a minimum concentration above 3 mg/m$^3$, and the maximum concentrations are well above 7 mg/m$^3$, reaching, in some cases, values of 12 mg/m$^3$ in the east transect. These plots coincide with areas where macroalgae were observed by locals and during in situ campaigns. Considering that the minimum values in these areas are well above the average in situ value measured on that day in the lagoon, it can be concluded that these concentrations are directly linked to the presence of macroalgae. This result is useful in the following sections, as it demonstrates that OC3 can be used as guidance for cluster estimation.

*3.2. Clustering*

Considering the atmospherically corrected Sentinel-2 data, as presented in previous sections, and the different chlorophyll algorithms applied to each image, we then proceeded to apply a clustering algorithm. A Cascade Simple K-means algorithm was applied to different images to extract the optimal number of clusters that represent areas with maximum differences between them. At the same time, each cluster is composed of pixels that share the maximum amount of similarities. The algorithm selected 9 clusters as the optimal amount in every image. Combinations of bands were tested to check which grouping produces clusters that could be good representations of the algal locations. Many different combinations were tested, but here, we introduce the three combinations that are most relevant to our discussion and that led to the optimal result. These are presented in Figure 4 for the image captured on 10 May 2022. The first combination included all Sentinel-2 spectral bands, OC3 and NDCI. The second combination included all Sentinel-2 spectral bands, OC3, and chl_mishra. The third combination included all Sentinel-2 spectral bands, OC3, NDCI, and chl_mishra.

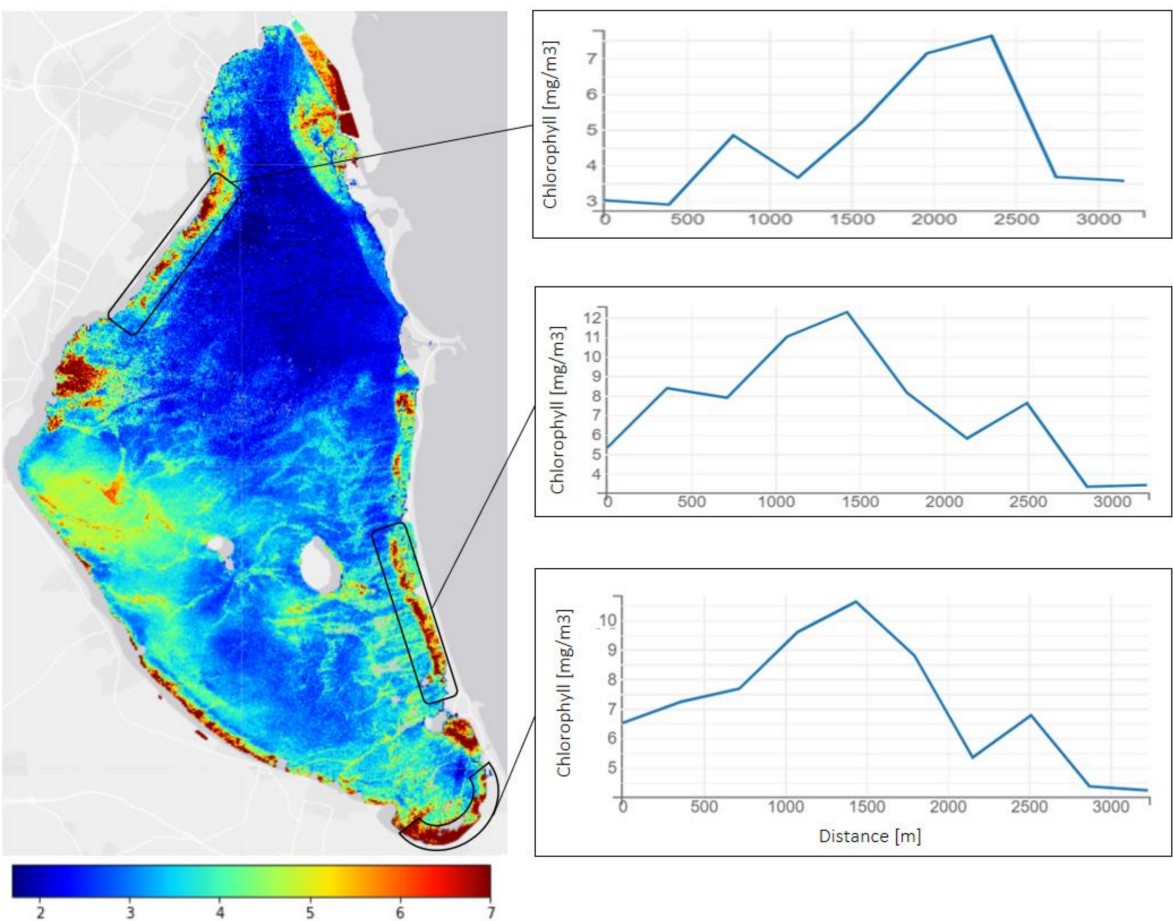

**Figure 3.** Map showing chlorophyll-a (mg/m$^3$) from OC3 on 10 May 2022; the gradient goes from 1.7 mg/m$^3$ (blue) to 7 mg/m$^3$ (red). Transects are plotted in areas of interest. **Top** panel: Northwest transect parallel to the coast of Los Alcázares. **Middle** panel: East transect parallel to the coast of La Manga. **Bottom** panel: South transect parallel to the coast east of Playa Honda.

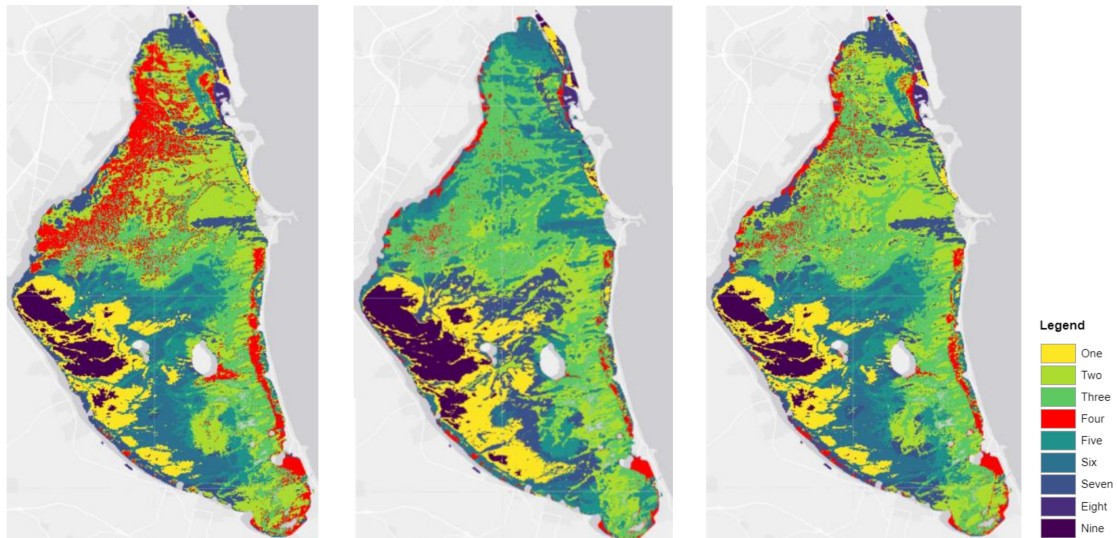

**Figure 4.** Clustering result in image from 10 May. Nine clusters selected as optimal number. The red cluster represents macroalgae based on in situ expert information. **Left** panel: all spectral bands, OC3 and NDCI. **Central** panel: all spectral bands, OC3 and chl_mishra. **Right** panel: all spectral bands, OC3, chl_mishra, and NDCI (this is the one selected to carry forward).

The cluster that most likely represents the macroalgae has been depicted in red in Figure 4. While the three combinations show similar clusters one to three and five to nine, cluster four (the macroalgae cluster) slightly varies depending on the variables included in the algorithm. For example, by not including chl_mishra (left panel), cluster four expands, and by not including NDCI (central panel), the size of the potential algal cluster is reduced. The option with all spectral bands, OC3, chl_mishra, and NDCI (right panel), was chosen to be the most representative of the actual locations of algae, based on the effect of band selection on the expected outcome, and how this matches the in situ experience.

As described in the methodology, these clusters are taken as a "synthetic in situ dataset" representative of the macroalgae locations. In the next step, these clusters are isolated into two types: "macroalgae" and "non-macroalgae" (all other clusters), and the information on the cluster that each pixel belongs to is added as a band to the satellite image. Finally, 5000 pixels (about 0.3% of the initial dataset) are randomly selected. Please note that these 5000 pixels are randomly extracted from both the "macroalgae" and "non-macroalgae" clusters, which means that, given the amount of "macroalgae" cluster present in each image, most of the pixels belong to the "non-macroalgae" cluster. This is positive, as it makes the classification exercise more complex and robust. These pixels constitute the training dataset together with the multispectral information for the classification algorithm.

### 3.3. Classification

A CART is now applied to the dataset obtained from the previous section (atmospherically corrected Sentinel-2 multispectral bands, chlorophyll indices, and cluster information). The classification exercise is applied to the images from 25 April, 10 May, and 25 May 2022 together, in order to increase the generalization capability of the resulting algorithm. The CART is trained with 5000 pixels from each image at once, and then the results are tested in unseen pixels of each image. The results of the classification algorithm are presented in Figure 5, top panels. The results of the classification have to be tested on pixels that have not been used to train the algorithm in order to assess its capability to produce accurate estimates. The classification results in those pixels are compared with the clustering results in the same pixels. Another new 5000 pixels per image are randomly selected for this part of the exercise, which we call the test dataset. The different metrics presented in previous sections are obtained for these pixels, i.e., accuracy, MCC, and confusion matrix. For the training dataset, all these metrics are 1, demonstrating perfect performance.

For the test dataset, the results are presented in Figure 5, bottom panels. The results show a constant accuracy of around 98%, while MCC ranges increase from 78% on 25 April to 86% on 25 May 2022. As described in previous sections, considering that the classes in this study are imbalanced (there are many more pixels in the no-macroalgae class than in the macroalgae class), the MCC is a much more representative metric than the accuracy. While the confusion matrix shows very consistent results for the model predictions for the "no macroalgae" class, the "macroalgae" class true positives decrease with time (90% for 25 April compared to 80% for 25 May). The results of the "no macroalgae" class are not surprising, considering that most of the pixels used to train the CART belong to this class, and, thus, the algorithm has much more information to draw a conclusion from. Although the results are very good, the model is less capable of accurately predicting the macroalgae class as time progresses. This could be due to the distribution of macroalgae being less well-defined in the image of 10 May, and of much lesser extent in the image of 25 May compared to 25 April, where the clusters of macroalgae were evenly distributed and well-defined. The use of these metrics helps us identify potential shortcomings of the classification algorithm and areas of improvement. However, when comparing clusters with classes with the naked eye, it is impossible to discern between them. This is reassuring from the point of view of zonification of macroalgal blooms and supports the use of these algorithms as a management tool for the lagoon.

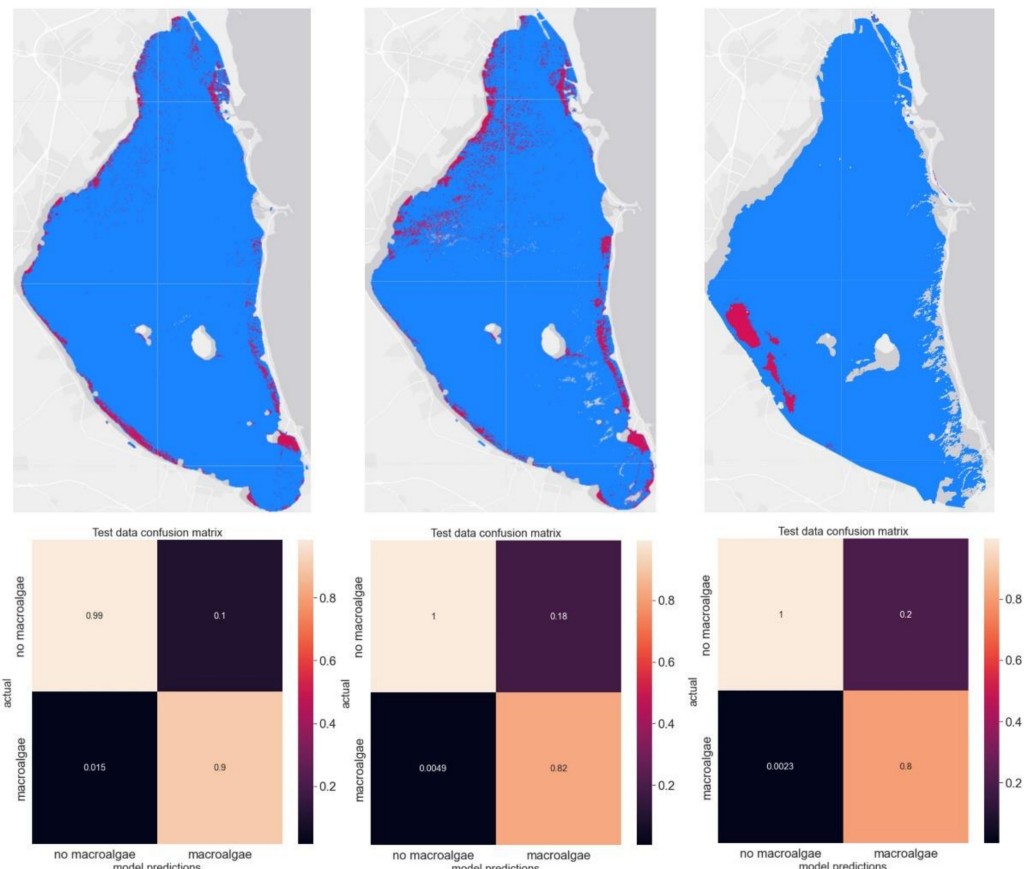

**Figure 5. Top** panels: classification results. 25 April 2022 (**left**), 10 May 2022 (**center**), 25 May 2022 (**right**). Pink: "macroalgae" class, blue: "no-macroalgae" class. **Bottom** panels: Confusion matrices for test dataset on 5000 random pixels (not used for training) extracted from images on 25 April 2022 (**left**, accuracy = 0.98213, MCC score = 0.78551), 10 May 2022 (**center**, accuracy = 0.98406, MCC score = 0.86200), and 25 May 2022 (**right**, accuracy = 0.98647, MCC score = 0.86796). The color bar represents the percentage of pixels in a given category.

### 3.4. Algorithm Explainability: SHAP Values to Understand Input Relevance in Output

SHAP values are a method based on cooperative game theory and are used to increase the transparency and interpretability of machine learning models [30]. This is a useful way to increase the expandability of artificial intelligence models and obtain information about what parameters are most representative of the overall outcome of the training exercise. This methodology discloses the individual contribution of each input variable to the output of the model for each observation. We can then extract useful statistics to understand general trends. This is particularly useful for developing semi-analytical models based on the outputs of machine learning. The SHAP values for the classification exercise are presented in Figure 6. The panel on the left shows a summary distribution of all SHAP values for all observations of the classification. This figure is very useful for understanding the range of variation of each parameter weight, but also to see the type of correlation (positive being red and negative being blue). The variables are ordered from most relevant (top) to least relevant (bottom). It is interesting to observe that NDCI and chl_mishra [20] are the most relevant inputs, ahead of any reflectance bands. Both present a positive contribution to the output. The result makes sense, as these are both chlorophyll indicators that already represent complex band relationships. NDCI is a normalized difference of the bands at 708 (close to Sentinel-2 red edge band B5) and 665 nm (Sentinel-2 red band B4). However, chl_mishra is a measure of chlorophyll, instead of an index. Therefore, NDCI may be considered a more robust index to guide machine learning algorithms, considering that it is normalized and within a given range, a fact that is useful for comparing different

images. Blue, aerosols (coastal blue), red edge, green, SWIR1, and chlorophyll from OC3 are the next most representative parameters. It is interesting to note that several spectral bands are more relevant for the results than OC3 chlorophyll. This is most likely because NDCI provides a good indicator that overcomes any potential improvement by OC3. Given the nature of the situation in the lagoon, NDCI also seems more relevant as it was initially developed for turbid waters. The panel on the right shows the mean SHAP values for all observations (that is, the mean of the distribution shown in the left panel). We can see how the influence of NDCI, chl_mishra, and B2 (band blue) is more significant than those of other variables.

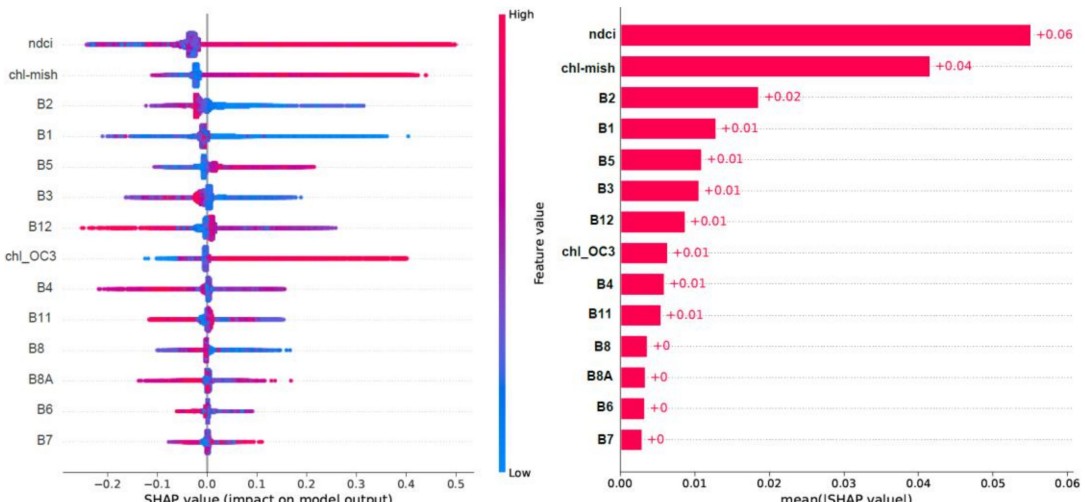

**Figure 6.** Summary SHapley Additive exPlanations (SHAP) values (**left**) and mean of absolute SHAP values (**right**).

### 3.5. Probability of a Macroalgal Bloom

The classification carried out in previous sections can be approached from a probabilistic point of view. In this case, instead of obtaining a binary classification as the outcome of the model, we obtain a set of probabilities ranging from 0 (total certainty that it is "no macroalgae" class) to 1 (total certainty that it is "macroalgae" class). In the classification algorithm, the values are normally approximated to the closest binary values for simplification (as, in many situations, the user is only interested in the class that a given pixel is closest to). However, the classifier output can be specified to provide probabilistic values instead. The gradient values between 0 and 1 provide an estimation of how close a pixel is to a given class, or what the probability is of a pixel being algae or not. These probabilities are useful for better understanding the limitations of the machine learning algorithm, but also provide certainty on the values that definitely belong to a given class. The results of this exercise are presented in Figure 7, where purple represents probability = 0 ("no macroalgae" class) and red represents probability = 1 (macroalgae class) on the image of 25 April 2022. Three areas of interest have been highlighted for further discussion. In those areas, we can see how some pixels show colors in between the purple-red maximum range, representing a degree of uncertainty in the mapping of the class that the pixels belong to.

It is of particular relevance to highlight the results presented in the right panel in Figure 7: there are a couple of straight lines that expand from the top to the bottom of the image. These striping effects are linked to the presence of scan lines in the original Sentinel-2 image due to the parallax effect between odd and even detectors (different detector footprints acquired from slightly different viewing angles). Therefore, these are anomalies in the image, and they present a good opportunity to check the performance of the algorithm. The probability in the lines is around 0.2, demonstrating that the algorithm has identified that the characteristics in those pixels are not equal to those in water, but they are definitely not algae. This type of result provides confidence when judging the

capabilities of the algorithm, and it is of particular interest when thinking of management strategies and zonification. Environmental managers can use probability maps to focus on areas where the presence of macroalgae is more certain in order to concentrate mitigation and removal strategies.

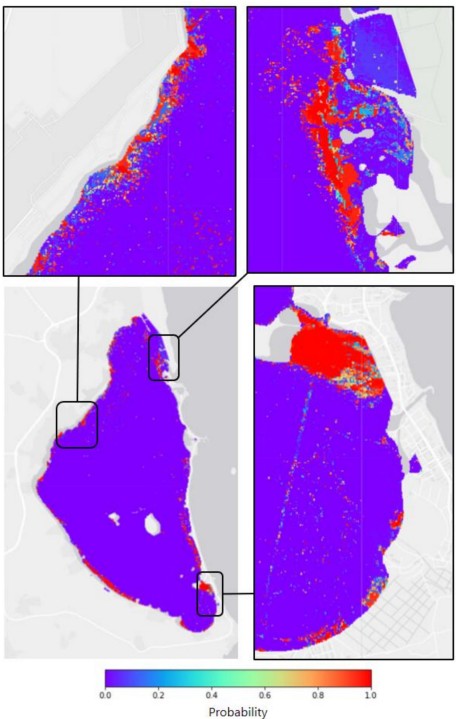

**Figure 7.** Probability of classification on 25 April 2022. Probability is used as an indicator of confidence on the prediction made by the classification algorithm presented in the previous section. The probability color scale ranges from 0 in purple ("no macroalgae" class) to 1 in red ("macroalgae" class).

*3.6. Prediction of Future Macroalgal Blooms*

As a final test, the algorithm was evaluated on images that were not previously used for training. In this case, an image from 4 June 2022 was provided, only using the patterns learned from the images in April and May, as discussed previously in the paper. Without any other information from the test image in June, the algorithm performed poorly, not being able to provide accurate results. In order to evaluate the ability of the algorithm, we decided to provide a minimal amount of information from the image in June, as specified in the table in Figure 8: a certain number of pixels from the new image were provided to capture the reflectance conditions on the day. One of the issues present when training the algorithm with the proposed methodology is that only a small amount of information about algae is provided, as the process is randomized. This means most of the data provided comes from the "no macroalgae" or water class.

Figure 8 presents the results of this exercise. In particular, the panel on the left shows the original clusters, while the panel in the center and right show the performance of the classification algorithm including 47 pixels from the unseen image from June (center) and 244 pixels (right). Of those 47 and 244 pixels, only 2 and 12 represent the macroalgae, respectively. It is clear that by providing just 2 algae locations to the algorithm, we reach a likelihood of about 50% of accuracy in the estimation. However, by including the slightly larger number of algal pixels, i.e., 12, we reach an 85% likelihood of determining the algal bloom correctly. Moreover, the right panel shows how a zonification can be accurate enough to have a good estimate of the macroalgal locations, i.e., when giving the algorithm between 2 and 12 algae locations. This can be easily carried out by recording in situ observations prior to evaluating the algorithm on the new satellite image.

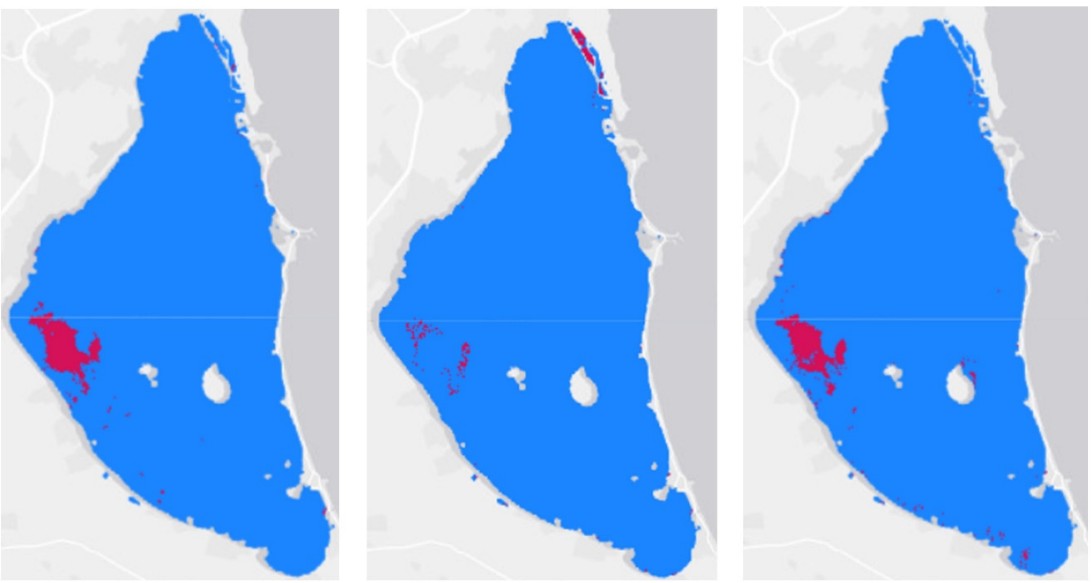

| Total Number of Points from 4 June 2022 Image | Number of Points in "Macroalgae" Cluster | Macroalgae Predicts Macroalgae (True Positive) | Macroalgae Predicts No Macroalgae (False Negative) | Accuracy | MCC Score |
|---|---|---|---|---|---|
| 47 | 2 (4.3%) | 0.55 | 0.45 | 0.95187 | 0.36717 |
| 244 | 12 (4.9%) | 0.85 | 0.15 | 0.99037 | 0.80485 |

**Figure 8.** "Macroalgae-no macroalgae" original clusters on 4 June 2022 (**left** panel, highlighted); results of classification on image from 4 June 2022 including 47 random points from new image in training (**center** panel); results of classification on image from 4 June 2022 including 244 random points from new image in training (**right** panel). Bottom: summary of metrics for 4 June 2022 image as evaluation image, including a given number of pixels from that image together with 5000 pixels from each of the images from 25 April 2022, 10 May 2022, and 25 May 2022. "True negatives" and "False positives" are above 0.99 and below 0.01, respectively, in all cases and have not been included in the table.

In summary, the performance of the algorithm in images previously unknown to it needs a baseline that determines the reflectance conditions on the day in order to tune the results accurately. Although to start with, we rely on having a minimal amount of in situ feedback to perform accurate estimates with this methodology, it is important to note that only three images have been used for this analysis, providing a very limited pool for the machine learning techniques to learn about the field. The more information that is provided to the algorithm, the lower the dependency on in situ observations. This is a very important point for future research in the area: the more information that is recorded about the presence of macroalgal blooms, the better this type of automatic detection technique can be tuned, helping create a database of conditions throughout the year that will help future predictions be less reliant on in situ observation, thus reducing management costs. Sentinel-2 satellites can certainly record the reflectance spectra that contain the scattering, absorption, and fluorescence signatures of algal blooms near the ocean surface [12–16,31], providing a 5-day revisit at the Equator (improved for higher latitudes) of global coastal zones.

## 4. Conclusions

This study demonstrates that the combined use of Sentinel-2 satellite imagery, Artificial Intelligence, and Machine Learning provided a robust tool for near-real-time detection of the appearance of macroalgae blooms in the Mar Menor coastal lagoon. The methodology can be used in near-real-time as an early warning system during future events. It facilitates

the work of the competent administrations to remove macroalgae with directed and non-random activities, as well as saving time and costs of monitoring vessels. In 2022, more than 17,000 Tons of this macroalgae were removed from the lagoon over 8 months. The strategy is beneficial for multiple socioeconomic sectors, such as tourism (by removing the macroalgae early on, it gives a better image of the beaches and the water of the lagoon) or fishing (the removal of the macroalgae prevents the fishing nets from collapsing and allows fishermen to carry out their work). In addition, it has been shown that macroalgae absorb large amounts of nutrients, such as nitrates and phosphates, which harm the lagoon, so their removal implies the elimination of a large amount of these elements from the waters. The current methodology can be easily transferred to other coastal regions with similar environmental issues. These guidelines target coastal managers in government and scientific researchers, for translating satellite remote sensing into information and tools that are useful for monitoring eutrophication and macroalgal blooms, which represents one of the most severe and widespread coastal environmental problems related to climate change. The encouraging values added by the satellite products in terms of synoptic observations and frequency are of paramount significance for ecological and crisis management purposes at local and regional scales. However, more information is required in order to define the optimal automatic detection techniques, helping create a database of conditions throughout the year that will help future predictions be less reliant on in situ observation, reducing management costs.

**Author Contributions:** Conceptualization, I.C. and G.N.; formal analysis and investigation, E.M.-L. and I.C.; original draft preparation, E.M.-L. and I.C.; writing—review and editing, J.S.-E., P.B. and G.N.; fieldwork, J.S.-E. and P.B; supervision, I.C.; project administration, I.C. and G.N; funding acquisition, I.C., J.S.-E. and G.N. All authors have read and agreed to the published version of the manuscript.

**Funding:** The research was supported by the Regional Government of Andalusia (SAT4ALGAE project, PY20-00244). This research was also funded by grants RTI2018-098784-J-I00 (Sen2Coast Project) and IJC2019-039382-I (Juan de la Cierva-Incorporación) funded by MCIN/AEI/10.13039/ 501100011033 and by "ERDF A way of making Europe". This work was carried out within the SEEME project (PID2019-109355RA-I00, MICIU/AEI/FEDER, EU) supported by the Spanish State Research Agency (AEI) and the Ministry of Science, Innovation and Universities (MICIU). The Scottish Funding Council and the Marine Alliance for Science and Technology for Scotland (MASTS) funded the research for this paper developed by E. Medina-Lopez through the Saltire Emerging Researcher Scheme, 2022. This work represents a contribution to the CSIC Thematic Interdisciplinary Platform PTI TELEDETECT.

**Data Availability Statement:** Satellite-derived and in situ data that support the findings of this study can be provided upon request.

**Acknowledgments:** We would like to acknowledge the European Commission's Copernicus program and the European Space Agency for distributing Sentinel-2 and imagery. The authors would like to thank Martha B. Dunbar for her English revision.

**Conflicts of Interest:** The authors declare no conflict of interest.

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
