# Peer review of "Machine Learning for Detection of Macroalgal Blooms in the Mar Menor Coastal Lagoon Using Sentinel-2"

_remotesensing, doi:10.3390/rs15051208_

Round 1

Reviewer 1 Report

1: The grammar needs minor corrections, additionally, I noticed a few spelling mistakes. I suggest using a tool like Grammarly to rectify the English.

2: The abstract is too long, and disjoined. I have provided a more succinic abstract at the end of the document. The same could be said for much of the article but I have focused on the science rather than the grammar.
3: Your images were not attached to my review copy.
4: You need to explain what you did in more detail (general comment)

4a-page 3, line 112. How large is the image (pixels and distance)
4b-pg3, li 145 show each of these equations

4c pg 4 li 170 what clustering fields were used li 175 what exactly were the 16 layers of information?

4d pg 4 li 197 what is the WEKA project, what is the Google Earth Engine?

4e pg 4 li 217 explain SVM and CART in your introduction

4f li 248, check formatting gap in equation

4g pg 7 li 332 typo "of: inserted into "accuracy"

4h pg 8, li 376 other factors, water depth, waves/stagnant/moving conditions not considered?

4i pg 9, li 414 history provided details of where it has accumulated in the past, perhaps a monthly summary would be a good input field?

pg 9 li 453 comma or dot? 17 tons or 17 kilotons?
The introduction needs revision to be more succinct

Also references 2, 3, and 4 appear overused and badly presented. I would

suggest making each of 2,3, and 4 into a single paragraph summarizing each article

Abstract too long. Shorten. Suggestion below

The Mar Menor coastal lagoon in Spain has experienced a decline in water quality due to increased nutrient input, leading to the eutrophication of the lagoon and the occurrence of microalgal and macroalgal blooms. In this study, machine learning techniques were applied to satellite imagery to identify the locations of macroalgae within the lagoon and understand the likelihood of future blooms. In-situ observations were also used to support the analysis. The results showed that the methodology was able to accurately identify the locations of macroalgae with accuracies above 98% and Matthew's Correlation Coefficients above 78%. The analysis also identified the key parameters contributing to the classification of pixels as algae, which could be used to develop future algorithms for detecting macroalgal blooms. This information can be used by environmental managers to implement early warning and mitigation strategies to prevent water quality deterioration in the lagoon. The usefulness of satellite observations for ecological and crisis management at local and regional scales is also highlighted.

Author Response

REVIEWER 1

Comments and Suggestions for Authors

1: The grammar needs minor corrections, additionally, I noticed a few spelling mistakes. I suggest using a tool like Grammarly to rectify the English.

The authors would like to thank the anonymous reviewer for providing useful and constructive comments that demonstrate a significant effort in reviewing the manuscript. We have carefully read all the annotations and enhanced the manuscript based on these minor suggestions. We have assessed and acknowledged all the points of the reviewer as described below. These changes are also identified in the revised manuscript using track changes. In addition, following the reviewer’s suggestion, we asked a native colleague to correct the spelling mistakes. The current version of the revised manuscript significantly benefitted from the reviewer’s insightful comments.

2: The abstract is too long, and disjoined. I have provided a more succinic abstract at the end of the document. The same could be said for much of the article but I have focused on the science rather than the grammar.

Response: the abstract has been shortened, thank you for the suggestion made.

3: Your images were not attached to my review copy.

Response: sorry for this issue, the images were in another document. The current revised document included all the figures.

4: You need to explain what you did in more detail (general comment)

Response: Thanks for this comment. We added addition information on the main methodology.

4a-page 3, line 112. How large is the image (pixels and distance)

Response: the following has been added to the text:” Each Sentinel-2 image cropped around the lagoon is several gigabytes in size and con-tains more than 1.5 million pixels.”

4c pg 4 li 170 what clustering fields were used

Response: the clustering exercise provides as a result a set of clusters that have certain characteristics in common. These characteristics are unknown, as it is not the purpose of the clustering exercise to provide this information. The clusters that represent algal blooms are chosen based on the location of in situ samples. These then feed into a classification exercise and SHAP analysis that then allows us to extract information on what fields represent each cluster. We have included a better explanation in the text to make this clearer.

li 175 what exactly were the 16 layers of information?

Response: the 16 layers are specified after the colon, but we have included a more detailed description for clarity.

4d pg 4 li 197 what is the WEKA project, what is the Google Earth Engine?

Response: these have been now explained in the text.

4e pg 4 li 217 explain SVM and CART in your introduction

Response: we have decided not to include SVM, given it has not been used to present our results. We have now included CART and k-Means in our introduction.

4f li 248, check formatting gap in equation

Response: this has been fixed.

4g pg 7 li 332 typo "of: inserted into "accuracy"

Response: this has been fixed.

4h pg 8, li 376 other factors, water depth, waves/stagnant/moving conditions not considered?

Response: no other conditions have been considered, as the goal of the paper is to be able to identify algal blooms only from satellite-derived information.

4i pg 9, li 414 history provided details of where it has accumulated in the past, perhaps a monthly summary would be a good input field?

Response: Unfortunately, we do not have a monthly summary as this would imply a great effort and cost. However, as explained in the text, we have compiled information on the places with the presence of macroalgae in different months. These data come from two sampling campaigns carried out by ourselves, campaigns from other IEO-CSIC colleagues, data provided by different fishermen's associations and, finally data collected from the written press.

Please find in the table below a summary of all these data with the date and the location of the macroalgae blooms:

Date

Latitude (N)

Longitude

Source

3rd February 2020

37,818518

-0,780308

Newspapers

14th February 2020

37,818518

-0,780308

Newspapers

25th May 2020

37,818518

-0,780308

Fishermen

2nd April 2020

37,818518

-0,780308

Newspapers

19th April 2020

37,818518

-0,780308

Fishermen

5th July 2021

37,692149

-0,835999

Sampling campaign

27th February 2022

37,653302

-0,787118

Sampling campaign

23rd April 2022

37,695776

-0,749627

Fishermen

23rd April 2022

37,686675

-0,746365

Newspapers

3rd May 2022

37,676429

-0,824669

Sampling campaign

11th May 2022

37,712459

-0,7555

Sampling campaign

16th May 2022

37,645719

-0,766202

Sampling campaign

19th May 2022

37,693689

-0,746672

Sampling campaign

23rd May 2022

37,642472

-0,746724

Sampling campaign

25th May 2022

37,652301

-0,786721

Sampling campaign

25th May 2022

37,650738

-0,781142

Sampling campaign

25th May 2022

37,692027

-0,835934

Sampling campaign

25th May 2022

37,634358

-0,730684

Sampling campaign

This table has been added in the manuscript.

In fact, one of the strengths of this study is to obtain the location of the macroalgae without having to carry out a visual reconnaissance of the entire lagoon, which means a great saving of time and effort.

pg 9 li 453 comma or dot? 17 tons or 17 kilotons?

Response: this has been fixed.

The introduction needs revision to be more succinct

Response: The authors agree that the Introduction section is succinct with the current topic of study.

Also references 2, 3, and 4 appear overused and badly presented. I would suggest making each of 2,3, and 4 into a single paragraph summarizing each article

Response: we have described each of the articles in the revised version of the manuscript.

Abstract too long. Shorten. Suggestion below

The Mar Menor coastal lagoon in Spain has experienced a decline in water quality due to increased nutrient input, leading to the eutrophication of the lagoon and the occurrence of microalgal and macroalgal blooms. In this study, machine learning techniques were applied to satellite imagery to identify the locations of macroalgae within the lagoon and understand the likelihood of future blooms. In-situ observations were also used to support the analysis. The results showed that the methodology was able to accurately identify the locations of macroalgae with accuracies above 98% and Matthew's Correlation Coefficients above 78%. The analysis also identified the key parameters contributing to the classification of pixels as algae, which could be used to develop future algorithms for detecting macroalgal blooms. This information can be used by environmental managers to implement early warning and mitigation strategies to prevent water quality deterioration in the lagoon. The usefulness of satellite observations for ecological and crisis management at local and regional scales is also highlighted.

Response: the abstract has been shortened.

Reviewer 2 Report

The MS investigated the satellite derived maroalgal blooms in Mar Menor Lagoon by using a set of machine learning techniques. The author validated his classification, and suggested the accuarcy was above 98%. However, I found that the author also declare that the accuarcy was not that  so accurate through line 331-349 . That made me not convinced of the accuracy description in the abstract, and if it is not so accurate, I cannot find any merits compared with other semi-analytical techniques. It is also recommended to compare the products quality between field and machine learnning, which I cannot find through MS. some minor points are below:

1. line 131-139:  It is hard to know how the authors retrive the field data only through some date information. how to identify the algal bloom, how to collect the sample data? how to analyze the sample data? some detail information is necessary 

Author Response

REVIEWER 2

The MS investigated the satellite derived maroalgal blooms in Mar Menor Lagoon by using a set of machine learning techniques.

The authors would like to thank the anonymous reviewer for providing useful and constructive comments that demonstrate a significant effort in reviewing the manuscript. We have carefully read all the annotations and enhanced the manuscript based on these minor suggestions. We have assessed and acknowledged all the points of the reviewer as described below. These changes are also identified in the revised manuscript using track changes. In addition, following one of the reviewer’s suggestions, we asked a colleague to correct the spelling mistakes. The current version of the revised manuscript significantly benefitted from the reviewer’s insightful comments.

The author validated his classification, and suggested the accuarcy was above 98%. However, I found that the author also declare that the accuarcy was not that so accurate through line 331-349 . That made me not convinced of the accuracy description in the abstract, and if it is not so accurate, I cannot find any merits compared with other semi-analytical techniques.

Response: please note that in the abstract we refer to accuracy and MCC score. While accuracy (defined in section 2.3.3 as the ratio between correctly predicted instances and all instances) is very high, MCC score (defined in section 2.3.3. as a Pearson correlation related metric, taking into account all instances, both correctly and incorrectly predicted) provides more realistic values of what is a very complex problem. Accuracy alone is not a reliable indicator of any prediction methodology, and thus we believed only providing that metric is not realistic. Unless other studies also provide MCC scores, it is difficult to compare actual results. All of the accuracies presented in our paper are above 98%. At any rate, the goal of this paper is to introduce a methodology to estimate blooms only from satellite imagery in a particular study area, which has not been presented in any previous studies, and the possibilities of this analysis for the study area.

It is also recommended to compare the products quality between field and machine learnning, which I cannot find through MS. some minor points are below:

  1. line 131-139:  It is hard to know how the authors retrive the field data only through some date information. how to identify the algal bloom, how to collect the sample data? how to analyze the sample data? some detail information is necessary how to analyze the sample data? some detail information is necessary 

Response: The information on the places with the presence of macroalgae in different months has been compiled. These data come from two sampling campaigns carried out by ourselves, campaigns from other IEO-CSIC colleagues, data provided by different fishermen's associations and finally data collected from the written press.

In fact, one of the strengths of this study is to obtain the location of the macroalgae without having to carry out a visual reconnaissance of the entire lagoon, which means a great saving of time and effort.

Some detailed information has been added within the text in the new versión.

For clarification, a summary of all these data with the date and the location of the macroalgae blooms is provided in the next table:

Date

Latitude (N)

Longitude

Source

3rd February 2020

37,818518

-0,780308

Newspapers

14th February 2020

37,818518

-0,780308

Newspapers

25th May 2020

37,818518

-0,780308

Fishermen

2nd April 2020

37,818518

-0,780308

Newspapers

19th April 2020

37,818518

-0,780308

Fishermen

5th July 2021

37,692149

-0,835999

Sampling campaign

27th February 2022

37,653302

-0,787118

Sampling campaign

23rd April 2022

37,695776

-0,749627

Fishermen

23rd April 2022

37,686675

-0,746365

Newspapers

3rd May 2022

37,676429

-0,824669

Sampling campaign

11th May 2022

37,712459

-0,7555

Sampling campaign

16th May 2022

37,645719

-0,766202

Sampling campaign

19th May 2022

37,693689

-0,746672

Sampling campaign

23rd May 2022

37,642472

-0,746724

Sampling campaign

25th May 2022

37,652301

-0,786721

Sampling campaign

25th May 2022

37,650738

-0,781142

Sampling campaign

25th May 2022

37,692027

-0,835934

Sampling campaign

25th May 2022

37,634358

-0,730684

Sampling campaign

Reviewer 3 Report

The article describes a method to help monitor and mitigate the effects of a recent increase of eutrophication in the Mar Menor coastal lagoon in Spain. Machine learning techniques are applied to remote sensing data and products to develop an algorithm to identify the location of macroalgae blooms in the lagoon. The method is then analyzed to determine the importance of each input to the model.

Section 3.5 is unclear. The idea of using probabilities of detection rather than absolutes is very good, but the way the probabilities were determined is not given. Also, in the paragraph starting on line 442 it is not clear what inputs were used initially, what the ‘minimal amount of information from the image’ was exactly. It appears to be saying that in situ information about the location of, at least some, macroalgae is required. If so, that information should be included in the previous analysis of the importance of inputs. 

I think only one of the larger scale maps on the left of Figure 1 is needed. 

Lines 76-77: the units for the phytoplankton bloom – mg/m^3 of chla, C, ?

Figure 3, bottom panel on right: text (line 281) says playa Honda (as labeled in Figure 1) but the caption says Punta de las Lomas. Use the same label and make sure it is on the map.

Line 402: are these the mean SHAP values or the mean of the absolute values?

There are several English language and style revisions needed throughout the manuscript.

Author Response

REVIEWER 3

The article describes a method to help monitor and mitigate the effects of a recent increase of eutrophication in the Mar Menor coastal lagoon in Spain. Machine learning techniques are applied to remote sensing data and products to develop an algorithm to identify the location of macroalgae blooms in the lagoon. The method is then analyzed to determine the importance of each input to the model.

The authors would like to thank the anonymous reviewer for providing useful and constructive comments that demonstrate a significant effort in reviewing the manuscript. We have carefully read all the annotations and enhanced the manuscript based on these minor suggestions. We have assessed and acknowledged all the points of the reviewer as described below. These changes are also identified in the revised manuscript using track changes. In addition, following one of the reviewer’s suggestions, we asked a colleague to correct the spelling mistakes. The current version of the revised manuscript significantly benefitted from the reviewer’s insightful comments.

Section 3.5 is unclear. The idea of using probabilities of detection rather than absolutes is very good, but the way the probabilities were determined is not given.

Response: This has now been explained in section 3.5.

Also, in the paragraph starting on line 442 it is not clear what inputs were used initially, what the ‘minimal amount of information from the image’ was exactly. It appears to be saying that in situ information about the location of, at least some, macroalgae is required. If so, that information should be included in the previous analysis of the importance of inputs. 

Response: we have made this sentence clearer in the text. A new subsection heading has also been included to make clear the difference with previous sections.

I think only one of the larger scale maps on the left of Figure 1 is needed. 

Response: agreed and changed.

Lines 76-77: the units for the phytoplankton bloom – mg/m^3 of chla, C, ?

Response: Corrected in the revised version: “2021 up to a chlorophyll concentration of 20 mg/m3

Figure 3, bottom panel on right: text (line 281) says playa Honda (as labeled in Figure 1) but the caption says Punta de las Lomas. Use the same label and make sure it is on the map.

Response: this has been corrected in the caption of the figure.

Line 402: are these the mean SHAP values or the mean of the absolute values?

Response: it is mean absolute SHAP values, we have corrected this on the text.

There are several English language and style revisions needed throughout the manuscript.

Following one of the reviewer’s suggestions, we asked a colleague to correct the spelling mistakes.

Reviewer 4 Report

This manuscript presents an interesting study in detection of macroalgal blooms. It would be useful to coastal remote sensing community. The best part is two paragraphs and figure 8 on Page 13. I recommend this paper for publication.

Author Response

REVIEWER 4

This manuscript presents an interesting study in detection of macroalgal blooms. It would be useful to coastal remote sensing community. The best part is two paragraphs and figure 8 on Page 13. I recommend this paper for publication.

The authors would like to thank the anonymous reviewer for providing useful and constructive comments that demonstrate a significant effort in reviewing the manuscript. We have carefully read all the annotations and enhanced the manuscript based on these minor suggestions. We have assessed and acknowledged all the points of the reviewer as described below. These changes are also identified in the revised manuscript using track changes. In addition, following one of the reviewer’s suggestions, we asked a colleague to correct the spelling mistakes. The current version of the revised manuscript significantly benefitted from the reviewer’s insightful comments.

Round 2

Reviewer 2 Report

The authors made a significant improvements compared to previous version. Generally, the ms made the technique to public in details, which follow by the similar case. The ms added some figures such as the comparison with or without macroalgae to demonstrate the accuracy. I was satisfied with the revision made currently.  The paper would provide a good example for other case study. some minor mistakes should be avoided.

Line 297: one more space before  "the minimum"

Table 1: the font of table text should be same as other part in the text